# Relaxed Marginal Consistency for Differentially Private Query Answering

**Ryan McKenna, Siddhant Pradhan, Daniel Sheldon, Gerome Miklau**
College of Information and Computer Sciences
University of Massachusetts
Amherst, MA 01002
{rmckenna, sspradhan, sheldon, miklau}@cs.umass.edu

## Abstract

Many differentially private algorithms for answering database queries involve a step that reconstructs a discrete data distribution from noisy measurements. This provides consistent query answers and reduces error, but often requires space that grows exponentially with dimension. PRIVATE-PGM is a recent approach that uses graphical models to represent the data distribution, with complexity proportional to that of exact marginal inference in a graphical model with structure determined by the co-occurrence of variables in the noisy measurements. PRIVATE-PGM is highly scalable for sparse measurements, but may fail to run in high dimensions with dense measurements. We overcome the main scalability limitation of PRIVATE-PGM through a principled approach that relaxes consistency constraints in the estimation objective. Our new approach works with many existing private query answering algorithms and improves scalability or accuracy with no privacy cost.

## 1 Introduction

A central problem in the design of differentially private algorithms is answering sets of counting queries from a database. Many proposed algorithms follow the select-measure-reconstruct paradigm: they *select* a set of measurement queries, they privately *measure* them (using Gaussian or Laplace noise addition), and then they *reconstruct* the data or query answers from the noisy measurements. When done in a principled manner, the reconstruct phase serves a number of critical functions: it combines the noisy evidence provided by the measurement queries, it allows new unmeasured queries to be answered (with no additional privacy cost), and it resolves inconsistencies in the noisy measurements to produce consistent estimates, which often have lower error. In this paper, we propose a novel, scalable, and general-purpose approach to the reconstruct step. With a principled approach to this problem, future research can focus on the challenging open problem of query selection.

Most existing *general-purpose* methods for reconstruction cannot scale to high-dimensional data, as they operate over a vectorized representation of the data, whose size is exponential in the dimensionality [1–6]. Some special purpose methods exist that have better scalability, but are only applicable within a particular mechanism or in certain special cases [7–11, 5, 12–15]. A recently-proposed method, PRIVATE-PGM [16], offers the scalability of these special purpose methods and retains much of the generality of the general-purpose methods. PRIVATE-PGM can be used for the reconstruction phase whenever the measurements only depend on the data through its low-dimensional marginals. PRIVATE-PGM avoids the data vector representation in favor of a more compact graphical model representation, and was shown to dramatically improve the scalability of a number of popular mechanisms while also improving accuracy [16]. PRIVATE-PGM was used in the winning entry of the 2018 NIST differential privacy synthetic data contest [17, 18], as well as in *both* the first and second-place entry of the follow-up 2020 NIST differential privacy temporal map contest [19, 20].

35th Conference on Neural Information Processing Systems (NeurIPS 2021).

While PRIVATE-PGM is far more scalable than operating over a vector representation of the data, it is still limited. In particular, its required memory and runtime depend on the structure of the underlying graphical model, which in turn is determined by which marginals the mechanism depends on. When the mechanism depends on a modest number of carefully chosen marginals, PRIVATE-PGM is extremely efficient. But, as the number of required marginals increases, the underlying graphical model becomes intractably large, and PRIVATE-PGM eventually fails to run. This is due to the inherent hardness of exact marginal inference in a graphical model.

In this paper, we overcome the scalability limitations of PRIVATE-PGM by proposing a natural relaxation of the estimation objective that enforces specified *local* consistency constraints among marginals, instead of global ones, and can be solved efficiently. Our technical contributions may be of broader interest. We develop an efficient algorithm to solve a generic convex optimization problem over the local polytope of a graphical model, which uses a body of prior work on generalized belief propagation [21–31] and can scale to problems with millions of optimization variables. We also propose a variational approach to predict "out-of-model" marginals given estimated pseudo-marginals, which gives a completely variational formulation for both estimation and inference: the results are invariant to optimization details, including the approximate inference methods used as subroutines.

Our new approach, APPROX-PRIVATE-PGM (APPGM), offers many of the same benefits as PRIVATE-PGM, but can be deployed in far more settings, allowing effective reconstruction to be performed without imposing strict constraints on the selected measurements. We show that APPGM permits efficient reconstruction for HDMM [32], while also improving its accuracy, allows MWEM [5] to scale to far more measurements, and improves the accuracy of FEM [33].

## 2 Background

We first review background on our data model, marginals, and differential privacy, following [16].

**Data**  Our input data represents a population of individuals, each contributing a single record $\mathbf{x} = (x_1, \ldots, x_d)$ where $x_i$ is the $i^{th}$ attribute belonging to a discrete finite domain $\Omega_i$ of $n_i$ possible values. The full domain is $\Omega = \prod_{i=1}^{d} \Omega_i$ and its size $n = \prod_{i=1}^{d} n_i$ is exponential in the number of attributes. A dataset $\mathbf{X}$ consists of $m$ such records $\mathbf{X} = (\mathbf{x}^{(1)}, \ldots, \mathbf{x}^{(m)})$. It is often convenient to work with an alternate representation of $\mathbf{X}$: the *data vector* or *data distribution* $\mathbf{p}$ is a vector of length $n$, indexed by $\mathbf{x} \in \Omega$ such that $\mathbf{p}(\mathbf{x})$ counts the fraction of individuals with record equal to $\mathbf{x}$. That is, $\mathbf{p}(\mathbf{x}) = \frac{1}{m} \sum_{i=1}^{m} \mathbb{I}\{\mathbf{x}^{(i)} = \mathbf{x}\}, \forall \mathbf{x} \in \Omega$, where $\mathbb{I}\{\cdot\}$ is an indicator function.

**Marginals**  When dealing with high-dimensional data, it is common to work with *marginals* defined over a subset of attributes. Let $r \subseteq [d]$ be a *region* or *clique* that identifies a subset of attributes and, for $\mathbf{x} \in \Omega$, let $\mathbf{x}_r = (x_i)_{i \in r}$ be the sub-vector of $\mathbf{x}$ restricted to $r$. Then the marginal vector (or simply "marginal on $r$") $\boldsymbol{\mu}_r$, is defined by:

$$\boldsymbol{\mu}_r(\mathbf{x}_r) = \frac{1}{m} \sum_{i=1}^{m} \mathbb{I}\{\mathbf{x}_r^{(i)} = \mathbf{x}_r\}, \quad \forall \mathbf{x}_r \in \Omega_r := \prod_{i \in r} \Omega_i. \tag{1}$$

This marginal is the data vector on the sub-domain $\Omega_r$ corresponding to the attribute set $r$. Its size is $n_r := |\Omega_r| = \prod_{i \in r} n_i$, which is exponential in $|r|$ but may be considerably smaller than $n$. A marginal on $r$ can be computed from the full data vector or the marginal for any superset of attributes by summing over variables that are not in $r$. We denote these (linear) operations by $M_r$ and $P_{s \to r}$, so $\boldsymbol{\mu}_r = M_r \mathbf{p} = P_{s \to r} \boldsymbol{\mu}_s$ for any $r \subseteq s$. We will also consider vectors $\boldsymbol{\mu}$ that combine marginals for each region in a collection $\mathcal{C}$, and let $M_{\mathcal{C}}$ be the linear operator such that $\boldsymbol{\mu} = (\boldsymbol{\mu}_r)_{r \in \mathcal{C}} = M_{\mathcal{C}} \mathbf{p}$.

**Differential Privacy**  Differential privacy protects individuals by bounding the impact any one individual can have on the output of an algorithm.

**Definition 1** (Differential Privacy [34]). *A randomized algorithm $\mathcal{A}$ satisfies $(\epsilon, \delta)$-differential privacy if, for any input $\mathbf{X}$, any $\mathbf{X}' \in nbrs(\mathbf{X})$, and any subset of outputs $S \subseteq Range(\mathcal{A})$,*

$$\Pr[\mathcal{A}(\mathbf{X}) \in S] \leq \exp(\epsilon) \Pr[\mathcal{A}(\mathbf{X}') \in S] + \delta$$

Above, nbrs($\mathbf{X}$) denotes the set of datasets formed by replacing any $\mathbf{x}^{(i)} \in \mathbf{X}$ with an arbitrary new record $\mathbf{x}'^{(i)} \in \Omega$. When $\delta = 0$ we say $\mathcal{A}$ satisfies $\epsilon$-differential privacy.

# 3 Private-PGM

In this section we describe PRIVATE-PGM [16], a general-purpose reconstruction method applied to differentially private measurements of a discrete dataset. There are two steps to PRIVATE-PGM: (1) *estimate* a representation of the data distribution given noisy measurements, and (2) *infer* answers to new queries given the data distribution representation.

In particular, suppose an arbitrary $(\epsilon, \delta)$-differentially private algorithm $\mathcal{A}$ is run on a discrete dataset with data vector $\mathbf{p}_0$, where $\mathcal{A}$ only depends on $\mathbf{p}_0$ through its low-dimensional marginals $\boldsymbol{\mu}_0 = M_{\mathcal{C}} \mathbf{p}_0$ for some collection of cliques $\mathcal{C}$. The sample $\mathbf{y} \sim \mathcal{A}(\boldsymbol{\mu}_0)$ typically reveals noisy high-level aggregate information about the data. PRIVATE-PGM will first estimate a compact representation of a distribution $\hat{\mathbf{p}}$ that explains $\mathbf{y}$ well, and then answer new queries using $\hat{\mathbf{p}}$.

**Estimation: Finding a Data Distribution Representation**   PRIVATE-PGM first estimates a data vector by finding $\hat{\mathbf{p}}$ to solve the inverse problem $\min_{\mathbf{p}} L(M_{\mathcal{C}} \mathbf{p})$, where $L(\boldsymbol{\mu})$ is a convex loss function that measures how well $\boldsymbol{\mu}$ explains the observations $\mathbf{y}$. Since $L(M_{\mathcal{C}} \mathbf{p})$ only depends on $\mathbf{p}$ through its marginals, it is clear we can find the optimal *marginals* by instead solving the following problem.

**Problem 1** (Convex Optimization over the Marginal Polytope). *Given a clique set $\mathcal{C}$ and convex loss function $L(\boldsymbol{\mu})$, solve*

$$\hat{\boldsymbol{\mu}} \in \underset{\boldsymbol{\mu} \in \mathcal{M}(\mathcal{C})}{\arg\min} L(\boldsymbol{\mu}),$$

*where $\mathcal{M}(\mathcal{C}) = \{\boldsymbol{\mu} : \exists\, \mathbf{p}$ s.t. $M_{\mathcal{C}} \mathbf{p} = \boldsymbol{\mu}\}$ is the set of realizable marginals, known as the* marginal polytope *of $\mathcal{C}$ [35].*

The solution to this problem gives marginals that are *consistent* with some underlying data vector and, therefore, typically provide a better estimate of the true marginals than $\mathbf{y}$. In the general case, the loss function $L$ can simply be set to the *negative log likelihood*, i.e., $L(\boldsymbol{\mu}) = -\log \Pr[\mathcal{A}(\boldsymbol{\mu}) = \mathbf{y}]$,[1] however other choices are also possible. As a concrete motivating application, consider the case where the mechanism $\mathcal{A}$ adds Gaussian noise directly to the data marginals $\boldsymbol{\mu}_0$, i.e., $\mathcal{A}(\boldsymbol{\mu}_0) = \mathbf{y}$ where $\mathbf{y}_r = \boldsymbol{\mu}_{0,r} + \mathcal{N}(0, \sigma^2 I_{n_r})$. In this case, the log-likelihood is proportional to the squared Euclidean distance and gives the loss function $L(\boldsymbol{\mu}) = \|\boldsymbol{\mu} - \mathbf{y}\|_2^2$, so the problem at hand is an $L_2$ minimization problem. The theory for PRIVATE-PGM focuses on convex loss functions, but the algorithms are also used to seek local minima of Problem 1 when $L$ is non-convex.

**Graphical models**   Two remaining issues are how to solve Problem 1 and how to recover a full data vector from $\hat{\boldsymbol{\mu}}$. PRIVATE-PGM addresses both with *graphical models*. A graphical model with clique set $\mathcal{C}$ is a distribution over $\Omega$ where the unnormalized probability is a product of factors involving only subsets of variables, one for each clique in $\mathcal{C}$. It has the form

$$\mathbf{p}_{\boldsymbol{\theta}}(\mathbf{x}) = \frac{1}{Z} \exp\left(\sum_{r \in \mathcal{C}} \boldsymbol{\theta}_r(\mathbf{x}_r)\right).$$

---

**Algorithm 1** PROX-PGM [16]

**Input:** Convex loss function $L(\boldsymbol{\mu})$
**Output:** Marginals $\hat{\boldsymbol{\mu}}$, parameters $\hat{\boldsymbol{\theta}}$
$\hat{\boldsymbol{\theta}} = \mathbf{0}$
**for** $t = 1, \ldots, T$ **do**
    $\hat{\boldsymbol{\mu}} = \text{MARGINAL-ORACLE}(\hat{\boldsymbol{\theta}})$
    $\hat{\boldsymbol{\theta}} = \hat{\boldsymbol{\theta}} - \eta_t \nabla L(\hat{\boldsymbol{\mu}})$
**return** $\hat{\boldsymbol{\mu}}, \hat{\boldsymbol{\theta}}$

---

The real numbers $\boldsymbol{\theta}_r(\mathbf{x}_r)$ are *log-potentials* or *parameters*. The full parameter vector $\boldsymbol{\theta} = (\boldsymbol{\theta}_r(\mathbf{x}_r))_{r \in \mathcal{C}, \mathbf{x}_r \in \Omega_r}$ matches the marginal vector $\boldsymbol{\mu}$ in size and indexing, and the relationship between these two vectors is central to graphical models [35]:

- A parameter vector $\boldsymbol{\theta}$ determines a unique marginal vector $\boldsymbol{\mu}_{\boldsymbol{\theta}} \in \mathcal{M}(\mathcal{C})$, defined as $\boldsymbol{\mu}_{\boldsymbol{\theta}} = M_{\mathcal{C}} \mathbf{p}_{\boldsymbol{\theta}}$, the marginals of $\mathbf{p}_{\boldsymbol{\theta}}$. *Marginal inference* is the problem of (efficiently) computing $\boldsymbol{\mu}_{\boldsymbol{\theta}}$ from $\boldsymbol{\theta}$. It can be solved exactly by algorithms such as *variable elimination* or *belief propagation* with a junction tree [36]. We denote by MARGINAL-ORACLE an algorithm that outputs $\boldsymbol{\mu}_{\boldsymbol{\theta}}$ on input $\boldsymbol{\theta}$.

- For every $\boldsymbol{\mu} \in \mathcal{M}(\mathcal{C})$ with positive entries, there is a unique distribution $\mathbf{p}_{\boldsymbol{\theta}}$ in the family of graphical models with cliques $\mathcal{C}$ that has marginals $\boldsymbol{\mu}$, and $\mathbf{p}_{\boldsymbol{\theta}}$ has *maximum entropy among all distributions with marginals $\boldsymbol{\mu}$.*

---

[1] For mechanisms with continuous output values, interpret this as a negative log-density.

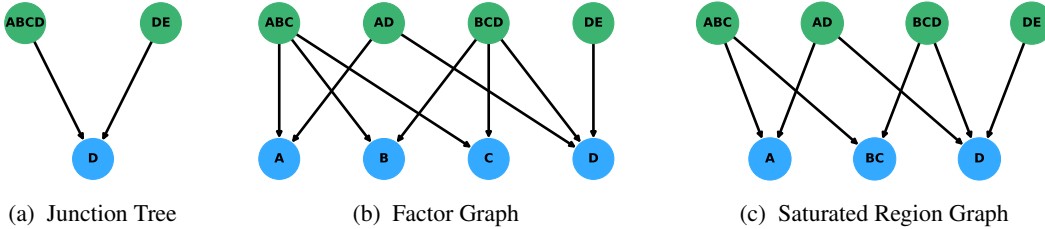

(a) Junction Tree        (b) Factor Graph        (c) Saturated Region Graph

Figure 1: Comparison of different region graph structures defined over a domain with attributes $\{A, B, C, D, E\}$ that support the cliques $\mathcal{C} = \{\{A, B, C\}, \{A, D\}, \{B, C, D\}, \{D, E\}\}$.

PROX-PGM (Algorithm 1) is a proximal algorithm to solve Problem 1 [16]. It returns a marginal vector $\hat{\boldsymbol{\mu}} \in \mathcal{M}(\mathcal{C})$ that minimizes $L(\boldsymbol{\mu})$ *and* parameters $\hat{\boldsymbol{\theta}}$ such that $\mathbf{p}_{\hat{\boldsymbol{\theta}}}$ has marginals $\hat{\boldsymbol{\mu}}$. The core of the computation is repeated calls to MARGINAL-ORACLE. The estimated graphical model $\mathbf{p}_{\hat{\boldsymbol{\theta}}}$ has cliques $\mathcal{C}$ that coincide with the marginals measured by the privacy mechanism, and has maximum entropy among all distributions whose marginals minimize $L(\boldsymbol{\mu})$. In general, there will be infinitely many distributions that have marginals $\hat{\boldsymbol{\mu}}$ (and hence are equally good from the perspective of the loss function $L$). PRIVATE-PGM chooses the distribution with maximum entropy, which is an appealing way to break ties that falls out naturally from the graphical model.

**Inference: Answering New Queries**    With $\mathbf{p}_{\hat{\boldsymbol{\theta}}}$ in hand, PRIVATE-PGM can readily estimate new marginals $\boldsymbol{\mu}_r$. There are two separate cases. If $r$ is contained in a clique of $C$, we say $r$ is "in-model", and we can readily calculate $\boldsymbol{\mu}_r$ from the output of PROX-PGM. The more interesting case occurs when $r$ is out-of-model (is not contained in a clique of $\mathcal{C}$): in this case, the standard way to compute $\boldsymbol{\mu}_r$ is to perform variable elimination in the graphical model $\mathbf{p}_{\boldsymbol{\theta}}$.

**Remark 1.** *The complexity of* PRIVATE-PGM *depends on that of* MARGINAL-ORACLE, *which depends critically on the structure of the cliques $\mathcal{C}$ measured by the privacy mechanism. In general, running time is exponential in the treewidth of the graph $G$ induced by attribute co-occurrence within a clique of $\mathcal{C}$. When $G$ is tree-like,* PRIVATE-PGM *can be highly efficient and exponentially faster than working with a full data vector. When $G$ is dense,* PRIVATE-PGM *may fail to run due to time or memory constraints. This limitation is not specific to the* PROX-PGM *algorithm: Problem 1 is as hard as marginal inference, which can be solved by minimizing the (convex) variational free energy over the marginal polytope [36]. A primary difficulty is the intractability of $\mathcal{M}(\mathcal{C})$, which is a convex set, but in general requires a very large number of constraints to represent explicitly [35]. In two state-of-the-art mechanisms for synthetic data, MST [18] and PrivMRF [20], $\mathcal{C}$ was specifically chosen to limit the treewidth and ensure tractability of Private-PGM. In other mechanisms agnostic to the limitations of* PRIVATE-PGM, *like HDMM and MWEM, the set $\mathcal{C}$ can often lead to graphs with intractable treewidths.*

## 4   Our Approach

In this section we describe our approach to overcome the main scalability limitations of PRIVATE-PGM, by introducing suitable approximations and new algorithmic techniques. Our innovations allow us to scale significantly better than PRIVATE-PGM with respect to the size of $\mathcal{C}$. A high-level idea is to use approximate marginal inference in the PROX-PGM algorithm, but doing so naively would make it unclear what, if any, formal problem is being solved. We will develop a principled approach that uses approximate inference to *exactly* solve a relaxed optimization problem.

**Region Graphs**    Central to our approach is the notion of a *region graph*, which is a data structure that encodes constraints between cliques in a natural graphical format and facilitates message passing algorithms for approximate marginal inference.

**Definition 2** (Region Graph [36]). *A region graph $G = (\mathcal{V}, \mathcal{E})$ is a directed graph where every vertex $r \in \mathcal{V}$ is an attribute clique and for any edge $r \rightarrow s \in \mathcal{E}$ we have that $r \supseteq s$. We say that $G$ supports a clique set $\mathcal{C}$ if for every clique $r \in \mathcal{C}$, there exists some $r' \in \mathcal{V}$ such that $r \subseteq r'$.*

For any region graph, there is a corresponding set of constraints that characterize the *local polytope* of internally consistent *pseudo-marginals*, defined below.

**Definition 3** (Local Polytope [36]). *The local polytope of pseudo-marginals associated with a region graph $G = (\mathcal{V}, \mathcal{E})$ is:*

$$\mathcal{L}(G) = \left\{ \boldsymbol{\tau} \geq 0 \ \middle| \ \begin{array}{ll} \mathbf{1}^{\top} \boldsymbol{\tau}_r = 1 & \forall r \in \mathcal{V} \\ P_{r \to s} \boldsymbol{\tau}_r = \boldsymbol{\tau}_s & \forall r \to s \in \mathcal{E} \end{array} \right\}. \tag{2}$$

The nodes in the region graph correspond to the cliques in the pseudo-marginal vector, while the edges in the region graph dictate which internal consistency constraints we expect to hold between two cliques. These constraints are necessary, but not sufficient, for a given set of pseudo-marginals to be realizable, i.e., $\mathcal{M}(\mathcal{V}) \subseteq \mathcal{L}(G)$. In the special case when $G$ is a junction tree, these constraints are also sufficient, and we have $\mathcal{M}(\mathcal{V}) = \mathcal{L}(G)$. We use the notation $\boldsymbol{\tau}$ in place of $\boldsymbol{\mu}$ to emphasize that $\boldsymbol{\tau}$ is not necessarily a valid marginal vector, even though we will generally treat it as such. This notational choice is standard in the graphical models literature [35]. The general idea is to relax problems involving the intractable marginal polytope to use the local polytope instead, since $\mathcal{L}(G)$ is straightforward to characterize using the linear constraints in Equation (2).

Region graphs can encode different structures, including junction trees and factor graphs as special cases. For example, Figure 1 shows three different region graphs that support $\mathcal{C} = \{\{A, B, C\}, \{A, D\}, \{B, C, D\}, \{D, E\}\}$. At one extreme is the Junction Tree, shown in Figure 1a, which is obtained by merging cliques $\{A, B, C\}$, $\{A, D\}$, and $\{B, C, D\}$ into a super-clique $\{A, B, C, D\}$. Here, $\mathcal{M}(\mathcal{V}) = \mathcal{L}(G)$, and ordinary belief propagation in this graph corresponds to exact marginal inference. At the other end of the extreme is the Factor Graph, shown in Figure 1b. This graph contains one vertex for every clique $r \in \mathcal{C}$, plus additional vertices for the singleton cliques. It encodes constraints that all cliques must agree on common one-way marginals. For example, $\boldsymbol{\tau}_{ABC}$ and $\boldsymbol{\tau}_{BCD}$ must agree on the shared one-way marginals $\boldsymbol{\tau}_B$ and $\boldsymbol{\tau}_C$, but not necessarily on the shared two-way marginal $\boldsymbol{\tau}_{BC}$. A natural middle ground is the fully Saturated Region Graph, shown in Figure 1c. This graph includes every clique $r \in \mathcal{C}$ as a vertex, and includes additional vertices to capture intersections between those cliques. Unlike the factor graph, this graph *does* require that $\boldsymbol{\tau}_{ABC}$ is consistent with $\boldsymbol{\tau}_{BCD}$ with respect to the $\boldsymbol{\tau}_{BC}$ marginal. Unlike the junction tree, this graph does not require forming super-cliques whose size grow quickly with $|\mathcal{C}|$. For more details about the concepts above, please refer to [36, Section 11.3].

The methods we describe in this paper apply for any region graph that supports $\mathcal{C}$. By default, we simply use the fully saturated region graph, which is the smallest region graph that encodes all internal consistency constraints, and can easily be constructed given the cliques $\mathcal{C}$ [36].

**Estimation: Finding an Approximate Data Distribution Representation** We begin by introducing a very natural relaxation of the problem we seek to solve.

**Problem 2** (Convex Optimization over the Local Polytope). *Given a region graph $G$ and a convex loss function $L(\boldsymbol{\tau})$ where $\boldsymbol{\tau} = (\boldsymbol{\tau}_r)_{r \in \mathcal{V}}$, solve:*

$$\hat{\boldsymbol{\tau}} = \underset{\boldsymbol{\tau} \in \mathcal{L}(G)}{\operatorname{argmin}} L(\boldsymbol{\tau}).$$

In the problem above, we simply replaced the marginal polytope $\mathcal{M}(\mathcal{V})$ from our original problem[2] with the local polytope $\mathcal{L}(G)$. Since this is a convex optimization problem with linear constraints, it can be solved with a number of general purpose techniques, including interior point and active set methods [37]. However, these methods do nothing to exploit the special structure in the constraint set $\mathcal{L}(G)$, and as such, they have trouble running on large-scale problems.

Our first contribution is to show that we can solve Problem 2 efficiently by instantiating PROX-PGM with a carefully chosen approximate marginal oracle. To do so, it is useful to view approximate marginal inference through the lens of the free energy minimization problem, stated below.

**Problem 3** (Approximate Free Energy Minimization [36]). *Let $G$ be a region graph, $\boldsymbol{\theta} = (\boldsymbol{\theta}_r)_{r \in \mathcal{V}}$ be real-valued parameters, and $H_\kappa(\boldsymbol{\tau}) = \sum_{r \in \mathcal{V}} \kappa_r H(\boldsymbol{\tau}_r)$, where $\kappa_r \in \mathbb{R}$ are counting numbers, and $H(\boldsymbol{\tau}_r) = -\sum_{\mathbf{x}_r \in \Omega_r} \boldsymbol{\tau}_r(\mathbf{x}_r) \log \boldsymbol{\tau}_r(\mathbf{x}_r)$ is the Shannon entropy of $\boldsymbol{\tau}_r$, solve:*

$$\hat{\boldsymbol{\tau}} = \underset{\boldsymbol{\tau} \in \mathcal{L}(G)}{\operatorname{argmin}} -\boldsymbol{\tau}^{\top} \boldsymbol{\theta} - H_\kappa(\boldsymbol{\tau})$$

---

[2]If $G$ supports $\mathcal{C}$, we can assume without loss of generality that Problem 1 was defined on $\mathcal{M}(\mathcal{V})$ instead of $\mathcal{M}(\mathcal{C})$. In particular, the loss function $L$ can be written to depend on marginals $(\boldsymbol{\mu}_{r'})_{r' \in \mathcal{V}}$ instead of $(\boldsymbol{\mu}_r)_{r \in \mathcal{C}}$, because the latter can be computed from the former.

This problem approximates the (intractable) variational free energy minimization problem [35], for which the optimum gives the true marginals of $\mathbf{p}_{\boldsymbol{\theta}}$, by using $\mathcal{L}(G)$ instead of $\mathcal{M}(\mathcal{V})$, and using $H_\kappa(\boldsymbol{\tau})$ as an approximation to the full Shannon entropy. Many algorithms for approximate marginal inference can be seen as solving variants of this free energy minimization problem under different assumptions about $G$ and $\kappa$ [21–31].

**Theorem 1** (Algorithm for Approximate Free Energy Minimization [25])**.** *Given a region graph $G$, parameters $\boldsymbol{\theta} = (\boldsymbol{\theta}_r)_{r\in\mathcal{V}}$, and any positive counting numbers $\kappa_r > 0$ for $r \in \mathcal{V}$, the convex generalized belief propagation (*CONVEX-GBP*) algorithm of [25] solves the approximate free energy minimization problem of Problem 3.*

CONVEX-GBP is listed in Appendix A (Algorithm 2) and is a message-passing algorithm in the region graph that uses the counting numbers as weights. Importantly, the complexity of CONVEX-GBP depends mainly on the size of the largest clique in the region graph. In many cases of practical interest, this will be exponentially smaller in the saturated region graph than in a junction tree.

**Theorem 2.** *When* PROX-PGM *uses* CONVEX-GBP *as the* MARGINAL-ORACLE *(with **any** positive counting numbers $\kappa$), it solves the convex optimization problem over the local polytope of Problem 2.*

This result is remarkable in light of previous work, where different counting number schemes are used with the goal of tightly approximating the true entropy, and form the basis for different approximate inference methods. In our setting, all methods with *convex* counting numbers are equivalent: they may lead to different *parameters* $\hat{\boldsymbol{\theta}}$, but the corresponding pseudo-marginals $\hat{\boldsymbol{\tau}} = $ CONVEX-GBP$(\hat{\boldsymbol{\theta}})$ are invariant. Indeed, the optimal $\hat{\boldsymbol{\tau}}$ depends only on the estimation objective $L(\boldsymbol{\tau})$ and the structure of the local polytope. We conjecture that a similar invariance holds for traditional marginal-based learning objectives with approximate inference [38] when message-passing algorithms based on convex free-energy approximations are used as the approximate inference method.

*Proof.* Since $L$ is a convex function and $\mathcal{L}$ is a convex constraint set, this problem can be solved with mirror descent [39]. Each iteration of mirror descent requires solving subproblems of the form:

$$\boldsymbol{\tau}^{t+1} = \underset{\boldsymbol{\tau}\in\mathcal{L}}{\operatorname{argmin}}\, \boldsymbol{\tau}^\top \nabla L(\boldsymbol{\tau}^t) + \frac{1}{\eta_t}D(\boldsymbol{\tau}, \boldsymbol{\tau}^t), \qquad D(\boldsymbol{\tau}, \boldsymbol{\tau}^t) = \psi(\boldsymbol{\tau}) - \psi(\boldsymbol{\tau}^t) - (\boldsymbol{\tau} - \boldsymbol{\tau}^t)^\top \nabla\psi(\boldsymbol{\tau}^t).$$

Here, $D$ is a Bregman distance measure and $\psi$ is some strongly convex and continuously differentiable function. Setting $\psi = -H_\kappa$, a negative weighted entropy with any (strongly) convex counting numbers $\kappa$, we arrive at the following update equation:

$$\begin{aligned}
\boldsymbol{\tau}^{t+1} &= \underset{\boldsymbol{\tau}\in\mathcal{L}(G)}{\operatorname{argmin}}\, \boldsymbol{\tau}^\top \nabla L(\boldsymbol{\tau}^t) + \frac{1}{\eta_t}\Big(-H_\kappa(\boldsymbol{\tau}) + H_\kappa(\boldsymbol{\tau}^t) + (\boldsymbol{\tau} - \boldsymbol{\tau}^t)^\top \nabla H_\kappa(\boldsymbol{\tau}^t)\Big) \\
&= \underset{\boldsymbol{\tau}\in\mathcal{L}(G)}{\operatorname{argmin}}\, \boldsymbol{\tau}^\top \Big(\eta_t \nabla L(\boldsymbol{\tau}^t) + \nabla H_\kappa(\boldsymbol{\tau}^t)\Big) - H_\kappa(\boldsymbol{\tau}) && \text{(algebraic manipulation)} \\
&= \underset{\boldsymbol{\tau}\in\mathcal{L}(G)}{\operatorname{argmin}}\, \boldsymbol{\tau}^\top \Big(\eta_t \nabla L(\boldsymbol{\tau}^t) - \boldsymbol{\theta}^t\Big) - H_\kappa(\boldsymbol{\tau}) && \text{(Lemma 1; Appendix A)} \\
&= \text{CONVEX-GBP}\big(G, \boldsymbol{\theta}^t - \eta_t \nabla L(\boldsymbol{\tau}^t), \kappa\big) && \text{(Theorem 1)}
\end{aligned}$$

$\square$

**Inference: Answering New Queries** We now turn our attention to the problem of inference. The central challenge is to estimate out-of-model marginals. Let $\hat{\boldsymbol{\tau}}$ and $\hat{\boldsymbol{\theta}}$ be the estimated pseudo-marginals and corresponding parameters after running PROX-PGM with CONVEX-GBP and region graph $G$. We have $\hat{\boldsymbol{\tau}} \approx \boldsymbol{\mu}_0$, and want an estimate $\hat{\boldsymbol{\tau}}_r \approx \boldsymbol{\mu}_{0,r}$ where $r \notin \mathcal{V}$.

CONVEX-GBP is the mapping such that $\hat{\boldsymbol{\tau}} = $ CONVEX-GBP$(\hat{\boldsymbol{\theta}}) \approx \boldsymbol{\mu}_0$. Thus, it is appropriate to use CONVEX-GBP with estimated parameters $\hat{\boldsymbol{\theta}}$ as the basis for estimating new pseudo-marginals. This requires selecting an expanded region graph $G'$ that supports $r$ and new counting number $\kappa_r$. In Appendix B, we analyze this approach for an idealized setting and find that it leads to estimates $\hat{\boldsymbol{\tau}}_r$ that maximize the entropy $H(\hat{\boldsymbol{\tau}}_r)$ subject to $\hat{\boldsymbol{\tau}}_r$ being consistent with $\hat{\boldsymbol{\tau}}$ on overlapping marginals. However, there are two practical difficulties with the idealized setting. First, there may be *no* $\hat{\boldsymbol{\tau}}_r$ that is consistent with $\hat{\boldsymbol{\tau}}$ on overlapping marginals: this is because $\hat{\boldsymbol{\tau}}$ satisfies only local consistency constraints. Second, the idealized case uses $\kappa_r$ very close to zero, and CONVEX-GBP performs poorly in this case. Instead, we design an optimization algorithm to mimic the idealized setting:

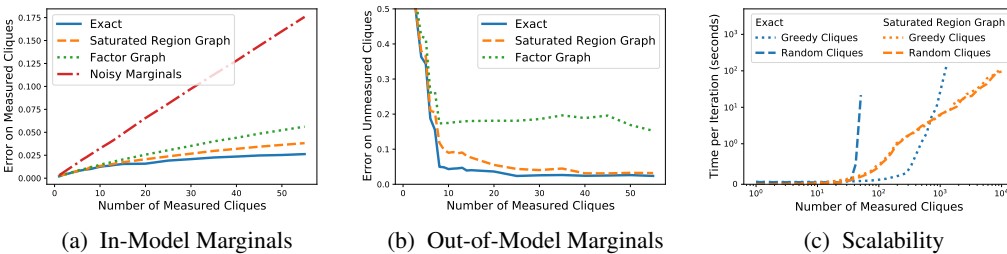

(a) In-Model Marginals        (b) Out-of-Model Marginals        (c) Scalability

Figure 2: Comparison between PROX-PGM with exact and approximate inference (for different region graph structures): (a) error on in-model (measured) marginals, (b) error on out-of-model (unmeasured) marginals, and (c) scalability of PROX-PGM vs. number of measured marginals.

**Problem 4** (Maximize Entropy Subject to Minimizing Constraint Violation). *Let $\hat{\boldsymbol{\tau}} = (\hat{\boldsymbol{\tau}}_u)_{u \in \mathcal{V}}$ and let $r \notin \mathcal{V}$. Solve:*

$$\max_{\hat{\boldsymbol{\tau}}_r} H(\hat{\boldsymbol{\tau}}_r) \text{ subject to } \hat{\boldsymbol{\tau}}_r \in \operatorname*{argmin}_{\boldsymbol{\tau}_r \in \mathcal{S}} \sum_{u \in \mathcal{V}, s = u \cap r} \|P_{r \to s} \boldsymbol{\tau}_r - P_{u \to s} \hat{\boldsymbol{\tau}}_u\|_2^2.$$

This relaxes the constraint that $\hat{\boldsymbol{\tau}}_r$ agrees with $\hat{\boldsymbol{\tau}}$ on overlapping marginals, to instead minimize the $L_2$ constraint violation. The inner problem is a quadratic minimization problem over the probability simplex $\mathcal{S}$. We show in Appendix B that a maximizer of Problem 4 is obtained by solving the inner problem once using entropic mirror descent [39].

The advantages of Problem 4 are that it is low-dimensional, only requires information from $\hat{\boldsymbol{\tau}}$ about attributes that are in $r$, can be solved much more quickly than running CONVEX-GBP, and can be solved in parallel for different marginals $r, r'$. This also gives a *fully* variational approach: both estimation and inference are fully defined through convex optimization problems that can be solved efficiently, and whose solutions are invariant to details of the approximate inference routines such as counting numbers.

## 5 Experiments

**Comparison to PRIVATE-PGM in a Simple Mechanism**    We begin by comparing the accuracy and scalability of APPGM and PRIVATE-PGM for estimating a fixed workload of marginal queries from noisy measurements of those marginals made by the Laplace mechanism. We use synthetic data to control the data domain and distribution (details in Appendix C.1) and measure $k$ different 3-way marginals with $\epsilon = 1$, for different settings of $k$. We run PROX-PGM with different versions of MARGINAL-ORACLE: Exact, Saturated Region Graph, and Factor Graph, where the first corresponds to PRIVATE-PGM, and the latter two to APPGM with the corresponding region graph (Figure 1).

*Accuracy.* We first show that when exact inference is tractable, some accuracy is lost by using approximate inference, but the estimated pseudo-marginals are much better than the noisy ones. We use eight-dimensional data with $n_1 = \cdots = n_8 = 4$, which is small enough so exact inference is always tractable, and measure random 3-way marginals for $k$ from 1 to $\binom{8}{3} = 56$. We then run 10000 iterations of PROX-PGM using differing choices for MARGINAL-ORACLE, and report the $L_1$ error on in- and out-of-model marginals, averaged over all cliques and across five trials.

In Figure 2a, we see that the error of all PROX-PGM variants is always lower than the error of the noisy measurements themselves. Exact (PRIVATE-PGM) always has lowest error, followed by Saturated Region Graph, then Factor Graph. This matches expectations: richer region graph structures encode more of the actual constraints and hence provide better error. The trend is similar for out-of-model marginals (Figure 2b). Factor Graph, which only enforces consistency with respect to the one-way marginals, performs poorly on unmeasured cliques, while Saturated Region Graph and Exact, which enforce more constraints, do substantially better. The difference between Saturated Region Graph and Exact is smaller, but meaningful.

*Scalability.* Next we consider high-dimensional data and compare the scalability of Exact and Saturated Region Graph on 100-dimensional data with $n_1 = \cdots = n_{100} = 10$. We vary $k$ from 1

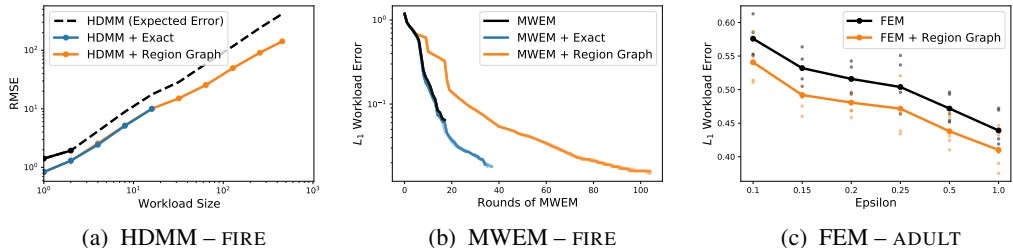

| (a) HDMM – FIRE | (b) MWEM – FIRE | (c) FEM – ADULT |

Figure 3: Three examples of using PROX-PGM to improve scalability and accuracy of other algorithms.

to $10^4$ and calculate the per-iteration time of PROX-PGM.[3] We consider two schemes for selecting measured cliques: *random* selects triples of attributes uniformly at random, and *greedy* selects triples to minimize the junction tree size in each iteration. As shown in Figure 2c, Exact can handle about 50 random measured cliques or 1000 greedy ones before the per-iteration time becomes too expensive (the growth rate is exponential). In contrast, Saturated Region Graph runs with 10000 measured cliques for either strategy and could run on larger cases (the growth rate is linear).

**Improving Scalability and Accuracy in Sophisticated Mechanisms**   We show next how PROX-PGM can be used to improve the performance of two sophisticated mechanisms for answering complex workloads of linear queries, including marginals. HDMM [32] is a state-of-the-art algorithm that first selects a "strategy" set of (weighted) marginal queries to be measured, and then reconstructs answers to workload queries. MWEM [5] is another competitive mechanism that iteratively measures poorly approximated queries to improve a data distribution approximation. In each algorithm, the bottleneck in high dimensions is estimating a data distribution $\hat{\mathbf{p}}$, which is prohibitive to do explicitly. PRIVATE-PGM can extend each algorithm to run in higher dimensions [16], but still becomes infeasible with enough dimensions and measurements. HDMM is a "batch" algorithm and either can or cannot run for a particular workload. Because MWEM iteratively selects measurements, even in high dimensions it can run for some number of iterations before the graphical model structure becomes too complex. By using APPGM instead, HDMM can run for workloads that were previously impossible, and MWEM can run for any number of rounds. We use the FIRE dataset from the 2018 NIST synthetic data competition [40], which includes 15 attributes and $m \approx 300{,}000$ individuals.

For HDMM, we consider workloads of $k$ random 3-way marginals, for $k = 1, 2, 4, 8, \ldots, 256, 455$, run five trials, and report root mean squared error, the objective function that HDMM optimizes. Figure 3a shows the results. HDMM with a full data vector cannot run for $k > 2$, but we can still analytically compute the *expected error* if it were able to run. HDMM + Exact fails to run beyond $k = 16$, while HDMM + Region Graph is able to run in every setting, substantially expanding the range of settings in which HDMM can be used. Both variants offer the significant error improvements of PGM-style inference, because they impose non-negativity constraints that reduce error. For example, when $k = 455$, there is a $3\times$ reduction in RMSE.

For MWEM, we consider the workload of all 2-way marginals, use a privacy budget of $\epsilon = 0.1$ *per round*, and run for as may rounds as possible, until MWEM has measured all 2-way marginals or exceeds a generous time/memory limit of 24 hours and 16GB. Figure 3b shows the results. As expected, MWEM + Exact runs successfully in early iterations, but exceeds resource limits by 35–40 rounds. In comparison, MWEM + Region Graph can run to completion and eventually measure all 2-way marginals. For a fixed number of rounds, Exact has lower error, in this case, substantially so, but results are data-dependent (we evaluate with other datasets in Appendix C). The difference can largely be traced to Exact's better performance on out-of-model cliques. In contrast, HDMM's measurements support all workload queries, so no out-of-model inference is required, and we see little gap between exact and approximate inference. Improving performance of approximate inference for out-of-model marginals is an area to be considered for future work; see Appendix B.

**Additional experiments**   We also apply APPGM to improve the accuracy of FEM, a recent state-of-the-art query-answering mechanism [33]. The setup is similar to the DualQuery experiment in [16]:

---

[3]Scalability experiments were conducted on two cores of a machine with a 2.4GHz CPU and 16 GB of RAM.

we run FEM to completion to release $\mathbf{y}$, but instead of answering queries directly with $\mathbf{y}$, we use APPGM to estimate pseudo-marginals, from which we compute query answers. This leads to a modest error reduction for all $\epsilon$ (Figure 3c; details in Appendix C). In Appendix C we also compare PRIVATE-PGM and APPGM directly to a method proposed in PriView [9] for estimating consistent marginals, and find that PGM-based methods are more accurate for almost all values of $\epsilon$. We additionally compare PRIVATE-PGM and APPGM to the recent Relaxed Projection method [14], and found that APPGM performs better for $\epsilon > 0.1$, although is outperformed for $\epsilon \leq 0.1$.

## 6 Practical Considerations and Limitations

When using our approach in practice, there are several implementation issues to consider. First note that CONVEX-GBP is an iterative algorithm that potentially requires many iterations to solve Problem 3, and this marginal inference routine is called within each iteration of PROX-PGM. Our theory requires running CONVEX-GBP until convergence, but in practice we only need to run it for a fixed number of iterations. In fact, we find that by *warm starting* the messages in CONVEX-GBP to the values from the previous call, we can actually run only one inner iteration of CONVEX-GBP within each outer iteration of PROX-PGM. This approach works remarkably well and makes much faster progress on reducing the objective than using more inner iterations. The main drawback of this is that we can no longer rely on a line search to find an appropriate step size within PROX-PGM, since one iteration of CONVEX-GBP with warm starting is not necessarily a descent direction. Using a constant step size works well most of the time, but selecting that step size can be tricky. We utilize a simple heuristic that seems to work well in most settings, but may require manual tuning in some cases. Second, we observed that introducing damping into CONVEX-GBP improved its stability and robustness, especially for very dense region graphs. Finally our MWEM experiment revealed that it is better to use PRIVATE-PGM over APPGM in the context of an MWEM-style algorithm, even though PRIVATE-PGM is more limited in the number of rounds it can run for. This performance difference can be traced back to our method for out-of-model inference, where utilization of local information only can lead to poor estimates. We discuss alternative approaches for this problem in Appendix B.

## 7 Related Work

A number of recent approaches support reconstruction for measurements on high-dimensional data. As part of the PriView algorithm [9], the authors describe a method for resolving inconsistencies in noisy measured marginals, which has since been incorporated into other mechanisms [41, 11, 42, 10]. Like APPGM, their method only guarantees local consistency. In Appendix C, we show empirically that it achieves similar (but slightly worse) error than APPGM. In addition, the method is less general, as measured queries may only be marginals, while APPGM allows an arbitrary convex loss function to be specified over the marginals. This extra generality is critical for integrating with mechanisms like HDMM, FEM, and MWEM when the workload contains more complex linear queries.

In concurrent work, Aydore et al. [14] and Liu et al. [15] proposed scalable instantiations of the MWEM algorithm, both avoiding the data vector representation in favor of novel compact representations. Although originally described in the context of MWEM-style algorithms, the key ideas presented in these works can be abstracted to the more general setting considered in this work. Specifically, these methods can be seen as alternatives to APPGM for overcoming the scalability limitations of PRIVATE-PGM; each of these methods make different approximations to overcome the inherent hardness of Problem 1. A direct comparison between PRIVATE-PGM or APPGM, and these alternatives remains an interesting question for future research.

To avoid the data vector representation, Liu et al. [13] restrict the support of the data vector to the domain elements that appear in a public dataset. This is much more scalable, but could result in poor performance if the public domain and the input domain differ substantially.

Lastly, Dwork et al. [12] propose an algorithm similar to APPGM. Their approach also projects onto an outer approximation of the marginal polytope, using the Frank Wolfe algorithm. The outer approximation is constructed via geometric techniques and is different from the local polytope we consider. They prove favorable error bounds with polynomial running time, but leave open the implementation and evaluation of their approach. By using the local polytope and message-passing algorithms, our method can scale in practice to problems with millions of variables.

## Acknowledgements

This work was supported by the National Science Foundation under grant IIS-1749854, by DARPA and SPAWAR under contract N66001-15-C-4067, and by Oracle Labs, part of Oracle America, through a gift to the University of Massachusetts Amherst in support of academic research.

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
