# A  Approximate Marginal Inference Algorithm

---

**Algorithm 2** CONVEX-GBP: Convex Generalized Belief Propagation [25]

---

**Input:** Region Graph $G = (\mathcal{V}, \mathcal{E})$, parameters $\boldsymbol{\theta} = (\boldsymbol{\theta}_r)_{r \in \mathcal{V}}$, convex counting numbers $\kappa_r > 0$
**Output:** Model marginals $\boldsymbol{\tau} = (\boldsymbol{\tau}_r)_{r \in \mathcal{C}}$
$\kappa_{r,t} = \kappa_r / (\kappa_t + \sum_{p \to t} \kappa_p)$
Initialize $m_{r \to t}(\mathbf{x}_t) = 0$ and $\lambda_{t \to r}(\mathbf{x}_t) = 0$
**for** $i = 1, \dots$ **do**
    **for** $r \to t$ **do**
        $m_{r \to t}(\mathbf{x}_t) = \kappa_r \log \left( \sum_{\mathbf{x}_r \setminus \mathbf{x}_t} \exp \left( (\boldsymbol{\theta}_r(\mathbf{x}_r) + \sum_{c \neq r} \lambda_{c \to r}(\mathbf{x}_c) - \sum_p \lambda_{r \to p}(\mathbf{x}_p))/\kappa_r \right) \right)$
        $\lambda_{t \to r}(\mathbf{x}_t) = \kappa_{r,t} \left( \boldsymbol{\theta}_t(\mathbf{x}_t) + \sum_c \lambda_{c \to t}(\mathbf{x}_t) + \sum_p m_{p \to t}(\mathbf{x}_t) \right) - m_{t \to r}(\mathbf{x}_t)$
    **for** $r \in \mathcal{C}$ **do**
        $\boldsymbol{\tau}_r(\mathbf{x}_r) \propto \exp \left( (\boldsymbol{\theta}_r(\mathbf{x}_r) + \sum_t \lambda_{t \to r}(\mathbf{x}_t) - \sum_p \lambda_{r \to p}(\mathbf{x}_r))/\kappa_r \right)$
**return** $\boldsymbol{\tau} = (\boldsymbol{\tau}_r)_{r \in \mathcal{V}}$

---

**Lemma 1.** *Let $G$ be a region graph, let $\kappa$ be positive counting numbers, and suppose $\hat{\boldsymbol{\tau}} = \text{CONVEX-GBP}(G, \hat{\boldsymbol{\theta}}, \kappa)$ for parameters $\hat{\boldsymbol{\theta}}$. Then, for any vector $\mathbf{z}$:*

$$\underset{\boldsymbol{\tau} \in \mathcal{L}(G)}{\arg\min} -\boldsymbol{\tau}^{\top} \left( -\nabla H_\kappa(\hat{\boldsymbol{\tau}}) + \mathbf{z} \right) - H_\kappa(\boldsymbol{\tau}) = \underset{\boldsymbol{\tau} \in \mathcal{L}(G)}{\arg\min} -\boldsymbol{\tau}^{\top} \left( \hat{\boldsymbol{\theta}} + \mathbf{z} \right) - H_\kappa(\boldsymbol{\tau})$$

For the remainder of this section, let $S$ be the linear subspace parallel to the affine hull of $\mathcal{L}(G)$, and let $S_\perp$ be the orthogonal complement of $S$. That is, if we write $\mathcal{L}(G) = \{\boldsymbol{\tau} \geq \mathbf{0} : A\boldsymbol{\tau} = \mathbf{b}\}$ using the constraint matrix $A$, then $S$ is the null space of $A$. This means that for any $\mathbf{0} < \boldsymbol{\tau} \in \mathcal{L}(G)$ and $\mathbf{z} \in S$, there is some $\lambda > 0$ such that $\boldsymbol{\tau} + \lambda\mathbf{z} \in \mathcal{L}(G)$. On the other hand, if $\mathbf{z} \in \mathcal{L}(G)$ and $\mathbf{z} \notin S$, there is no $\lambda > 0$ such that $\boldsymbol{\tau} + \lambda\mathbf{z} \in \mathcal{L}(G)$.

*Proof.* Because $\hat{\boldsymbol{\tau}} = \text{CONVEX-GBP}(G, \hat{\boldsymbol{\theta}}, \kappa)$ we know that $\hat{\boldsymbol{\tau}}$ minimizes $-\tilde{\boldsymbol{\tau}}^{\top} \hat{\boldsymbol{\theta}} - H_\kappa(\tilde{\boldsymbol{\tau}})$ over all $\tilde{\boldsymbol{\tau}} \in \mathcal{L}(G)$, and it is easy to see from the final line of CONVEX-GBP that $\hat{\boldsymbol{\tau}} > \mathbf{0}$. Therefore, we can apply Lemma 2 below to conclude that there are vectors $\mathbf{v}_\perp, \mathbf{v}'_\perp \in S_\perp$ such that

$$-\nabla H_\kappa(\hat{\boldsymbol{\tau}}) = \mathbf{u}(\hat{\boldsymbol{\tau}}) + \mathbf{v}_\perp$$

$$\hat{\boldsymbol{\theta}} = \mathbf{u}(\hat{\boldsymbol{\tau}}) + \mathbf{v}'_\perp$$

where $\mathbf{u}(\hat{\boldsymbol{\tau}})$ is the projection of $-\nabla H_\kappa(\hat{\boldsymbol{\tau}})$ onto $S$.

We will now show that the linear parts of the objectives of the two minimization problems in the lemma statement differ by only a constant. Since the nonlinear part is the same, this will prove that the objectives as a whole differ by only a constant, so the problems have the same minimizers, as stated.

Let $\mathbf{z} = \mathbf{z}_{\|} + \mathbf{z}_\perp$ where $\mathbf{z}_{\|} \in S$ and $\mathbf{z}_\perp \in S_\perp$. Then, for any $\boldsymbol{\tau} \in \mathcal{L}(G)$, the linear component of the first objective is

$$-\boldsymbol{\tau}^{\top}(-\nabla H_\kappa(\boldsymbol{\tau}) + \mathbf{z}) = -\boldsymbol{\tau}^{\top}(\mathbf{u}(\hat{\boldsymbol{\tau}}) + \mathbf{v}_\perp + \mathbf{z}_{\|} + \mathbf{z}_\perp) = -\boldsymbol{\tau}^{\top}(\mathbf{u}(\hat{\boldsymbol{\tau}}) + \mathbf{z}_{\|}) + c \qquad (3)$$

where $c = -\boldsymbol{\tau}^{\top}(\mathbf{v}_\perp + \mathbf{z}_\perp)$ is a constant that does not depend on $\boldsymbol{\tau}$, since, for any $\boldsymbol{\tau}, \boldsymbol{\tau}' \in \mathcal{L}(G)$ we have

$$\boldsymbol{\tau}'^{\top}(\mathbf{v}_\perp + \mathbf{z}_\perp) - \boldsymbol{\tau}^{\top}(\mathbf{v}_\perp + \mathbf{z}_\perp) = (\boldsymbol{\tau}' - \boldsymbol{\tau})^{\top}(\mathbf{v}_\perp + \mathbf{z}_\perp) = 0,$$

since $\boldsymbol{\tau}' - \boldsymbol{\tau} \in S$ and $\mathbf{v}_\perp + \mathbf{z}_\perp \in S_\perp$.

Similarly, the linear component of the second objective is

$$-\boldsymbol{\tau}^{\top}(\hat{\boldsymbol{\theta}} + \mathbf{z}) = -\boldsymbol{\tau}^{\top}(\mathbf{u}(\hat{\boldsymbol{\tau}}) + \mathbf{v}'_\perp + \mathbf{z}_{\|} + \mathbf{z}_\perp) = -\boldsymbol{\tau}^{\top}(\mathbf{u}(\hat{\boldsymbol{\tau}}) + \mathbf{z}_{\|}) + c' \qquad (4)$$

where $c' = -\boldsymbol{\tau}^T(\mathbf{v}'_\perp + \mathbf{z}_\perp)$ is a (different) constant independent of $\boldsymbol{\tau}$.

This shows that the objectives differ by a constant, and completes the proof.

Equation (3) and Equation (4) show that the objectives differ by a constant, which completes the proof. $\qquad \square$

**Lemma 2.** *Let $G$ be a region graph, let $\kappa$ be positive counting numbers, and let $\boldsymbol{\tau} \in \mathcal{L}(G)$ with $\boldsymbol{\tau} > 0$. Define $\Theta(\boldsymbol{\tau}) = \{\boldsymbol{\theta} : \boldsymbol{\tau} = \operatorname{argmin}_{\tilde{\boldsymbol{\tau}} \in \mathcal{L}(G)} -\tilde{\boldsymbol{\tau}}^\top \boldsymbol{\theta} - H_\kappa(\tilde{\boldsymbol{\tau}})\}$ to be the set of all $\boldsymbol{\theta}$ such that* CONVEX-GBP$(G, \boldsymbol{\theta}, \kappa) = \boldsymbol{\tau}$. *Then*

$$\Theta(\boldsymbol{\tau}) = \mathbf{u}(\boldsymbol{\tau}) + S_\perp$$

*where $\mathbf{u}(\boldsymbol{\tau})$ is the projection of $-\nabla H_\kappa(\boldsymbol{\tau})$ onto $S$.*

*Proof.* This follows fairly standard arguments in convex analysis after noting that the objective of the optimization problem coincides with the convex conjugate of $-H_\kappa$ (e.g., see Rockafellar, 2015[1]; Bertsekas, 2009[2]), but with some specialization to our setting.

Define $f(\boldsymbol{\tau})$ to be the extended real-valued function that takes value $-H_\kappa(\boldsymbol{\tau})$ for $\boldsymbol{\tau} \in \mathcal{L}(G)$ and $+\infty$ for $\boldsymbol{\tau} \notin \mathcal{L}(G)$. Let $\partial f(\boldsymbol{\tau})$ be the subdifferential of $f$ at $\boldsymbol{\tau} \in \mathcal{L}(G)$.

We will first show that $\Theta(\boldsymbol{\tau}) = \partial f(\boldsymbol{\tau})$.

By the definition of a subgradient, for $\boldsymbol{\tau} \in \mathcal{L}(G)$,

$$
\begin{aligned}
\boldsymbol{\theta} \in \partial f(\boldsymbol{\tau}) &\iff f(\tilde{\boldsymbol{\tau}}) \geq f(\boldsymbol{\tau}) + \boldsymbol{\theta}^\top (\tilde{\boldsymbol{\tau}} - \boldsymbol{\tau}) \quad \forall \tilde{\boldsymbol{\tau}} \in \mathbb{R}^d \\
&\iff f(\tilde{\boldsymbol{\tau}}) \geq f(\boldsymbol{\tau}) + \boldsymbol{\theta}^\top (\tilde{\boldsymbol{\tau}} - \boldsymbol{\tau}) \quad \forall \tilde{\boldsymbol{\tau}} \in \mathcal{L}(G) \\
&\iff \boldsymbol{\tau}^\top \boldsymbol{\theta} - f(\boldsymbol{\tau}) \geq \tilde{\boldsymbol{\tau}}^\top \boldsymbol{\theta} - f(\tilde{\boldsymbol{\tau}}) \quad \forall \tilde{\boldsymbol{\tau}} \in \mathcal{L}(G) \\
&\iff \boldsymbol{\tau} = \operatorname*{argmax}_{\tilde{\boldsymbol{\tau}} \in \mathcal{L}(G)} \tilde{\boldsymbol{\tau}}^\top \boldsymbol{\theta} - f(\tilde{\boldsymbol{\tau}}) \\
&\iff \boldsymbol{\tau} = \operatorname*{argmin}_{\tilde{\boldsymbol{\tau}} \in \mathcal{L}(G)} -\tilde{\boldsymbol{\tau}}^\top \boldsymbol{\theta} - H_\kappa(\tilde{\boldsymbol{\tau}}) \\
&\iff \boldsymbol{\theta} \in \Theta(\boldsymbol{\tau}).
\end{aligned}
$$

In the second line, we used the fact that the inequality always holds for $\tilde{\boldsymbol{\tau}} \notin \mathcal{L}(G)$ because $f(\tilde{\boldsymbol{\tau}}) = +\infty$ and the other quantities are finite. In the third line, we used the fact that $f(\tilde{\boldsymbol{\tau}})$, which coincides with $-H_\kappa(\tilde{\boldsymbol{\tau}})$ on $\mathcal{L}(G)$, is strictly convex, so $\boldsymbol{\tau}$ is a *unique* maximizer of $\tilde{\boldsymbol{\tau}}^\top \boldsymbol{\theta} - f(\tilde{\boldsymbol{\tau}})$.

Now, we will show that $\partial f(\boldsymbol{\tau}) = \mathbf{u}(\boldsymbol{\tau}) + S_\perp = \{\mathbf{u}(\boldsymbol{\tau}) + \mathbf{v} : \mathbf{v} \in S_\perp\}$, which will conclude the proof.

We use the following characterization of the subdifferential (Rockafellar, 2015, Theorem 23.2):

$$\boldsymbol{\theta} \in \partial f(\boldsymbol{\tau}) \iff \boldsymbol{\theta}^\top \mathbf{z} \leq f'(\boldsymbol{\tau}; \mathbf{z}) \quad \forall \mathbf{z} \in \mathbb{R}^d \tag{5}$$

where $f'(\boldsymbol{\tau}; \mathbf{z})$ is the directional derivative of $f$ along direction $\mathbf{z}$. Since $f$ is the restriction of the differentiable function $-H_\kappa$ to $\mathcal{L}(G)$, its directional derivatives coincide with those of $-H_\kappa$ for points $\boldsymbol{\tau} \in \mathcal{L}(G)$ with $\boldsymbol{\tau} > 0$ and directions in $\mathbf{z} \in S$ (so that $\boldsymbol{\tau} + \lambda \mathbf{z} \in \mathcal{L}(G)$ for small enough $\lambda > 0$), and are equal to $+\infty$ for points $\boldsymbol{\tau} \in \mathcal{L}(G)$ and directions $\mathbf{z} \notin S$ (so that $\boldsymbol{\tau} + \lambda \mathbf{z} \notin \mathcal{L}(G)$ for any $\lambda > 0$). That is, for $\boldsymbol{\tau} \in \mathcal{L}(G)$ with $\boldsymbol{\tau} > 0$,

$$f'(\boldsymbol{\tau}; \mathbf{z}) = \begin{cases} -\nabla H(\boldsymbol{\tau})^\top \mathbf{z} & \mathbf{z} \in S \\ +\infty & \mathbf{z} \notin S \end{cases} \tag{6}$$

Therefore, by Equation (5) and Equation (6), for any $\boldsymbol{\tau} \in \mathcal{L}(G)$ with $\boldsymbol{\tau} > 0$,

$$
\begin{aligned}
\boldsymbol{\theta} \in \partial f(\boldsymbol{\tau}) &\iff \boldsymbol{\theta}^\top \mathbf{z} \leq f'(\boldsymbol{\tau}; \mathbf{z}) && \forall \mathbf{z} \in \mathbb{R}^d \\
&\iff \boldsymbol{\theta}^\top \mathbf{z} \leq f'(\boldsymbol{\tau}; \mathbf{z}) && \forall \mathbf{z} \in S \\
&\iff \boldsymbol{\theta}^\top \mathbf{z} \leq -\nabla H(\boldsymbol{\tau})^\top \mathbf{z} && \forall \mathbf{z} \in S \\
&\iff \boldsymbol{\theta}^\top \mathbf{z} = -\nabla H(\boldsymbol{\tau})^\top \mathbf{z} && \forall \mathbf{z} \in S \\
&\iff \boldsymbol{\theta} = \mathbf{u}(\boldsymbol{\tau}) + \mathbf{v} && \mathbf{v} \in S_\perp,
\end{aligned}
$$

where $\mathbf{u}(\boldsymbol{\tau})$ is the projection of $-\nabla H(\boldsymbol{\tau})$ onto $S$. In the second line, we used the fact that the inequality always holds for $\mathbf{z} \notin S$ because $f'(\boldsymbol{\tau}; \mathbf{z}) = +\infty$ and the other quantities are finite. In the fourth line, we observed that $\mathbf{z} \in S$ iff $-\mathbf{z} \in S$ (since $S$ is a linear subspace) and

$$\boldsymbol{\theta}^\top(-\mathbf{z}) \leq -\nabla H(\boldsymbol{\tau})^\top(-\mathbf{z}) \iff \boldsymbol{\theta}^\top \mathbf{z} \geq -\nabla H(\boldsymbol{\tau})^\top \mathbf{z},$$

---

[1]Rockafellar, R. T. (2015). *Convex analysis*. Princeton university press.

[2]Bertsekas, Dimitri P. Convex optimization theory. Belmont: Athena Scientific, 2009.

so the third line is equivalent to both inequalities holding for all $\mathbf{z} \in S$. The equivalence of the final line to the penultimate line is a straightforward exercise by breaking both $\boldsymbol{\theta}$ and $-\nabla H(\boldsymbol{\tau})$ into their orthogonal components along $S$ and $S_\perp$, respectively, and observing that the component of $-\nabla H(\boldsymbol{\tau})$ along $S$ is $\mathbf{u}(\boldsymbol{\tau})$. $\qquad\square$

# B   Out-of-Model Inference

We now turn our attention to the problem of out-of-model inference; i.e., estimating $\boldsymbol{\tau}_r$ where $r \notin \mathcal{V}$. There are many approaches for this problem that seem natural on the surface, but upon close inspection each one has it's problems. In Section 4, we proposed one approach that had certain desirable properties, but we considered many alternatives which enumerate below and discuss in detail.

## B.1   Variable Elimination in $\mathbf{p}_{\boldsymbol{\theta}}$

In PRIVATE-PGM, out-of-model inference was done by performing variable elimination in the graphical model $\mathbf{p}_{\boldsymbol{\theta}}$. There are two problems with applying that idea here. First, variable elimination will not in general be tractable for the graphical models we may encounter, since it is an exact inference method. Second, if we run variable elimination to estimate in-model marginals from $\boldsymbol{\theta}$ produced by PROX-PGM, it will give a different answer than the pseudo-marginals $\boldsymbol{\tau}$ produced by PROX-PGM (even if $\boldsymbol{\tau} \in \mathcal{M}(\mathcal{V})$ is a realizable marginal). In this case, the pseudo-marginals estimated by PROX-PGM are the ones that should be trusted, and the parameters $\boldsymbol{\theta}$ are only useful in the context of our approximate marginal oracle CONVEX-GBP. In summary, this approach is not viable, and even if it was, it has undesirable properties.

Before moving on, we make note of an alternate way to perform exact out-of-model inference that will motivate our first approach to approximate out-of-model inference. They key idea is to add a new zero log-potentials $\boldsymbol{\theta}_r = \mathbf{0}$ for the new clique whose marginal we are interested in estimating. Clearly, the introduction of this zero log-potential does not change the distribution $\mathbf{p}_{\boldsymbol{\theta}}$ or it's in-model marginals $\boldsymbol{\mu}_{\boldsymbol{\theta}}$. However, when we run an exact MARGINAL-ORACLE with these new parameters, it will produce all in-model marginals, and the new out-of-model marginal as well.

## B.2   Running CONVEX-GBP on an Expanded Region Graph

Using the idea above, one approach to out-of-model inference is to expand the region graph to include the region $r$ whose pseudo-marginal we are interested in. This will require adding at least one new vertex $r$ to the region graph. Edges and additional vertices could be added depending on the structure of the existing region graph, and the desired local consistency constraints that $\boldsymbol{\tau}_r$ should obey. With this new region graph, we can set $\boldsymbol{\theta}_r = \mathbf{0}$ (and do the same for any additional vertices we added as well), and run CONVEX-GBP on the new graph. This is an interesting idea, but it leaves open several questions:

1. What nodes and edges should be included in the augmented region graph?

2. What counting numbers should be assigned to those nodes?

3. What formal guarantees can we make about this approach?

4. Can we analyze the message-passing equations in CONVEX-GBP to perform an equivalent computation without re-running the algorithm in its entirety?

For question (1) above, a natural choice is to use the same structure as the original region graph. For example, if the original region graph is a factor graph, then we can simply add one new vertex corresponding to the new one, and add edges connecting to the singleton cliques. If the original region graph is saturated, then we can build a new saturated region graph that includes the new clique.

For question (2) above, a natural choice is to use $\kappa'_r = 1$ for all regions $r$ (including the new region), since that is the scheme used to set $\kappa$ within PROX-PGM. Unfortunately, the new pseudo-marginals $\boldsymbol{\tau}' = \text{CONVEX-GBP}(\boldsymbol{\theta}', \kappa')$ may not agree with the originally optimized pseudo-marginals $\boldsymbol{\tau} = \text{CONVEX-GBP}(\boldsymbol{\theta}, \kappa)$ on the in-model cliques. Specifically, $\boldsymbol{\tau}_r$ need not equal $\boldsymbol{\tau}'_r$ when $r \in \mathcal{V}$. This is clearly undesirable, and would be a consistency violation. A better choice of the counting

numbers would be $\kappa'_r = \kappa_r$ for $r \in \mathcal{V}$ and $\kappa_{r'} = 0$ otherwise. As we show below, this approach has a compelling theoretical guarantee.

**Theorem 3.** *Let $G = (\mathcal{V}, \mathcal{E})$ be a region graph, $\boldsymbol{\theta}$ be parameters, $\kappa$ be positive counting numbers and let $\boldsymbol{\tau} = \text{CONVEX-GBP}(G, \boldsymbol{\theta}, \kappa)$. Now let $G' = (\mathcal{V}', \mathcal{E}')$ be a region graph that extends $G$ (i.e, $\mathcal{V} \subseteq \mathcal{V}'$ and $\mathcal{E} \subseteq \mathcal{E}'$), $\boldsymbol{\theta}'_r = \boldsymbol{\theta}_r$ if $r \in \mathcal{V}$ and $\boldsymbol{\theta}'_r = \mathbf{0}$ if $r \notin \mathcal{V}$, $\kappa'_r = \kappa_r$ for $r \in \mathcal{V}$ and $\kappa'_r = \varepsilon$ otherwise.*

$$\boldsymbol{\tau}' = \lim_{\varepsilon \to 0^+} \text{CONVEX-GBP}(G', \boldsymbol{\theta}', \kappa')$$

*If $S = \{\boldsymbol{\tau}' \in L(G') \mid \boldsymbol{\tau}'_r = \boldsymbol{\tau}_r \forall r \in \mathcal{V}\} \neq \varnothing$, then*

$$\boldsymbol{\tau}' = \operatorname*{argmax}_{\boldsymbol{\tau}' \in S} \sum_{r \in \mathcal{V}' \setminus \mathcal{V}} H(\boldsymbol{\tau}'_r)$$

*Proof.* We begin by restating the free energy minimization problem solved by CONVEX-GBP.

$$\begin{aligned}
\boldsymbol{\mu}' &= \operatorname*{argmin}_{\boldsymbol{\tau}' \in \mathcal{L}(G')} -\boldsymbol{\theta}^\top \boldsymbol{\tau}' - H_{\kappa'}(\boldsymbol{\tau}') \\
&= \operatorname*{argmin}_{\boldsymbol{\tau}' \in \mathcal{L}(G')} -\Big[ \sum_{r \in \mathcal{V}} \boldsymbol{\theta}_r^\top \boldsymbol{\tau}' + \kappa_r H(\boldsymbol{\tau}'_r) \Big] - \Big[ \sum_{r \in \mathcal{V}' \setminus \mathcal{V}} \mathbf{0}^\top \boldsymbol{\tau}' + \varepsilon H(\boldsymbol{\tau}'_r) \Big] \\
&= \operatorname*{argmin}_{\boldsymbol{\tau}' \in \mathcal{L}(G')} -\Big[ \sum_{r \in \mathcal{V}} \boldsymbol{\theta}_r^\top \boldsymbol{\tau}' + \kappa_r H(\boldsymbol{\tau}'_r) \Big] - \varepsilon \sum_{r \in \mathcal{V}' \setminus \mathcal{V}} H(\boldsymbol{\tau}'_r)
\end{aligned}$$

Note that as $\varepsilon \to 0$, the objective only depends on $\boldsymbol{\tau}_r$ for $r \in \mathcal{V}$ (and not $r \in \mathcal{V}' \setminus \mathcal{V}$). Thus, $\boldsymbol{\tau}_r$ only affect the problem via the constraints they impose on the problem. Since $\boldsymbol{\tau}$ is the optimizer of the relaxed problem when $L(G')$ is replaced by $L(G)$ (which includes a subset of the constraints), if $\boldsymbol{\tau}$ is feasible in the larger problem (which it is by assumption $S \neq \varnothing$), it is also optimal in this problem. Moreover, since we are taking the limit as $\varepsilon \to 0$ from the right, there will be an infinitesimally small entropy penalty, which will force $\boldsymbol{\mu}'_r$ to have maximum entropy among marginals that are consistent with $\boldsymbol{\mu}$, as desired. $\square$

Theorem 3 is a compelling reason to use this approach, namely running CONVEX-GBP with zero counting numbers for the new cliques whose marginals we are estimating. One subtle detail to this theorem is that it is certainly possible that $S = \varnothing$, which means that there aren't any pseudo-marginals in the expanded region graph that are consistent with the pseudo-marginals in the original region graph. In this case, it is not immediately clear how to characterize the behavior of this approach.

**Remark 2** (Special Case: Factor Graph). *In the special case when both the original and expanded region graphs are factor graphs, we can guarantee that $S \neq \varnothing$ and we can efficiently estimate the new pseudo-marginal without rerunning CONVEX-GBP over the full graph. Since factor graphs only require each pseudo-marginal to be internally consistent with respect to the one-way marginals, we can always find higher-order marginals by multiplying the one-way marginals. Clearly, this gives the maximum entropy estimate for the new pseudo-marginal that is internally consistent with the existing ones. This is a computationally cheap estimate: it simply requires multiplying one-way marginals and does not require any iterative message passing scheme.*

For more complex region graphs, things do not work out so nicely. Since we are mainly interested in saturated region graphs in this work, this nice result for factor graphs is not particularly useful for our purposes. In practice, there is a problem with running CONVEX-GBP with a zero or near-zero counting numbers. We observed empirically that using small counting numbers severely deteriorates the convergence rate of CONVEX-GBP, and for that reason, this is not an ideal approach.

### B.3 Minimizing Constraint Violation and Maximizing Entropy

While the method described above has some drawbacks in practice, the principles underlying the approach are still sound: namely, we should find the maximum entropy distribution for the new

marginal that is consistent with the existing marginals (for some natural notion of consistency). However, for complex region graphs, it is certainly possible that no such marginals exist. In that case, a natural alternative would be to find a pseudo-marginal that minimizes the constraint violation, and among all minimizers, has maximum entropy. This is the approach that we evaluated empirically, and described in Section 4.

It requires solving a quadratic minimization problem over the probability simplex. This problem can be readily solved with iterative proximal algorithms like entropic mirror descent [39]. Entropic mirror descent guarantees the solution found will have maximum entropy among all minimizers of the objective. Thus, when $S \neq \varnothing$, this method gives the same answer as Theorem 3. However, it is more general, and also does something principled when $S = \varnothing$. Additionally, this method does not require any information about attributes not in $r$, and even though it is an iterative algorithm, each iteration runs much faster than an iteration of CONVEX-GBP.

## B.4 Running PROX-PGM over expanded local polytope.

While the idea above is more principled than the alternatives that preceded it, it is still not ideal because it does not guarantee perfect consistency between the in-model pseudo-marginals and the out-of-model pseudo marginals. When perfect consistency is not achievable, it settles for minimizing the constraint violation.

We can overcome this limitation by running PROX-PGM on an *over-saturated* region graph. That is the region graph will contain vertices for every region necessary to define the loss function, *and* every region whose pseudo-marginal we are interested in estimated. The additional regions do not affect the loss function (the log-potentials will always remain $\mathbf{0}$), but it does impact the constraints. In particular, upon convergence, the estimated pseudo-marginals will all be locally consistent. This comes at a cost, however. Since the region graph contains more vertices and edges, each iteration of PROX-PGM requires more time, and the algorithm as a whole is slower. Whether it makes sense to use this strategy depends on how important perfect consistency is, as well as how many new marginals must be answered. In our empirical evaluation of this approach, we found that it did produce better estimates than the previous idea, but also took considerably longer.

## B.5 Incorporating Global Information

As we saw empirically in Section 5, our approach to out-of-model inference did not perform particularly well compared to the exact method used in PRIVATE-PGM. We conducted more experiments to verify this in Appendix C.2. In this subsection, we explore in greater detail why it did not perform well in all cases.

Consider a simple graphical model with cliques $\mathcal{C} = \{\{A, B\}, \{B, C\}\}$, and suppose that $A$ is highly correlated with $B$ and $B$ is highly correlated with $C$. Then clearly, $A$ and $C$ should also be highly correlated. When performing exact inference in this model, we preserve this correlation between $A$ and $C$. However, when we only require local consistency for the new clique, we will assume that $A$ and $C$ are independent, and lose the correlation between $A$ and $C$.

Note that all methods described thus far suffer from this problem, not just the one method we evaluated in this paper. To correctly preserve the correlation between $A$ and $C$, we would have to first estimate the $\{A, B, C\}$ marginal then derive the $\{A, C\}$ marginal from it. This could be accomplished by adding an $\{A, B, C\}$ region to the region graph and using any of the methods described above. In this toy problem, it is easy enough to do and feasible, but for larger region graphs, it is not immediately obvious how to generalize the idea.

Since exact marginal inference is not feasible for the graphs we are interested in, it is clear that we must make some approximation. It is not clear what the nature of the approximation should be, however. We showed that only using local information in the approximation has problems in some cases, and utilizing some global information may give better results in some cases. We leave this as an interesting open problem.

# C  Additional Experiments

## C.1  Synthetic Data used in Experiments

Given a domain size $(n_1, \ldots, n_d)$ and a number of records $m$, we generate synthetic data to use in experiments as follows:

1. Compute a random spanning tree of the complete graph with nodes $1, \ldots, d$. The edges in this tree will correspond to the cliques in our model.

2. For each edge $r$ in the tree, set $\boldsymbol{\theta}_r \sim N(0, \sigma^2)^{n_r}$. Here $\sigma$ is a "temperature" parameter that determines the strength of the parameters.

3. Sample $m$ records from the graphical model $\mathbf{p}_{\boldsymbol{\theta}}$.

## C.2  MWEM Experiments

In Figure 3b of Section 5 we observed that integrating APPGM into MWEM can enable the mechanism to run for more rounds, but the approximation resulted in much worse error for the same number of rounds. When run to completion, MWEM+APPGM did achieve lower error than the minimum error achieved by MWEM+PRIVATE-PGM, but it required running for $3\times$ as many rounds and thus spending $3\times$ as much privacy budget. In the figure below, we include additional lines for different privacy levels $\epsilon = 0.05, 0.1, 0.2$ *per round*. As shown in the figure, it would be better to run MWEM + Exact for 35 round at $\epsilon = 0.1$ than it would be to run MWEM + Region Graph for 70 rounds at $\epsilon = 0.05$.

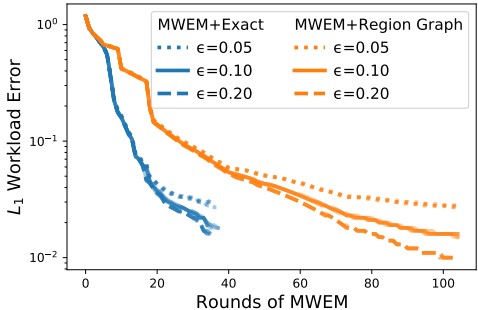

As hinted at in the main text, the main reason Region Graph performs poorly here is because it only incorporates local information when conducting out-of-model inference, which is problematic for this dataset. To demonstrate that this is really the problem, we repeat the experiment with $\epsilon = \infty$. That is, in each round of MWEM, we exactly select the worse approximated clique, and measure the corresponding marginal with no noise. Since no noise is added, the measured marginals solve Problem 2 and there is no need to run PROX-PGM. Thus, the only difference between Exact and Region Graph is in the handling of out-of-model marginals. We run the experiment for five datasets and plot the results below. The additional error for Region Graph is particularly large for the fire and msnbc dataset but not as much for adult, loans, and titanic. msnbc is a click stream dataset and is thus naturally modeled as a Markov chain. Once the 2-way marginals corresponding to the edges in this Markov chain are measured, MWEM + Exact achieves very low error. MWEM + Exact preserves the long range dependencies between the first and last node in the chain, whereas MWEM + Region Graph only preserves the local dependencies, which explains the difference in this case. Some datasets (like adult, loans, and titanic) do not have strong dependency chains as msnbc does, and in these cases there is a smaller difference in error for out-of-model marginals.

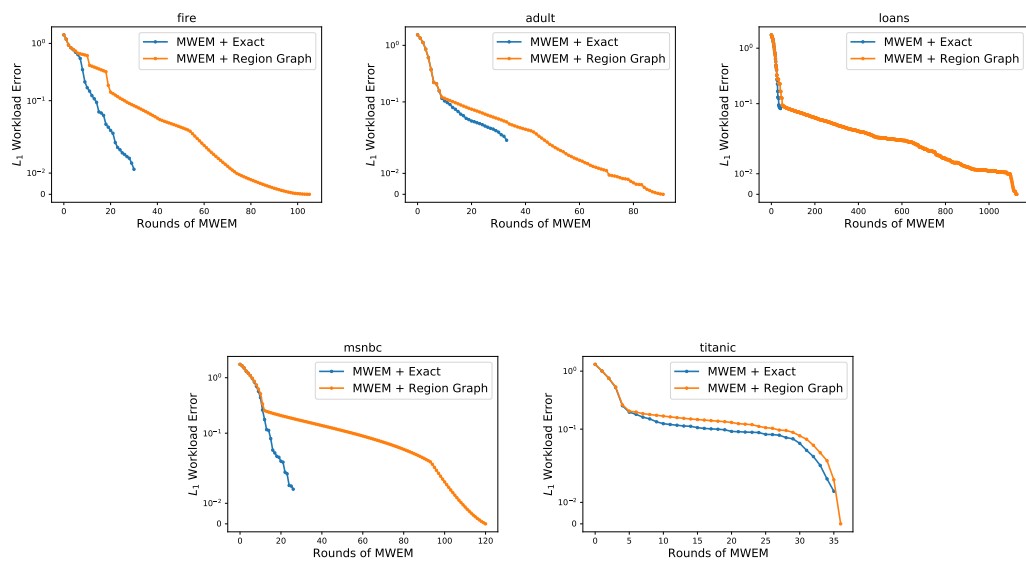

## C.3 FEM Experiments

As hinted at in Section 5, our method can be integrated into FEM as well [33]. The integration is similar to how PRIVATE-PGM is used to improve DualQuery [16, 43]. Like MWEM, FEM runs for a specified number of rounds, and maintains an estimate of the data distribution (in tabular format) throughout the execution. In each round, FEM selects a query from the workload that is poorly approximated under the current estimate of the data distribution using the exponential mechanism. This is the only way in which FEM interacts with the sensitive data (Unlike MWEM, it does not measure this query with Laplace noise). It then adds records to the estimated dataset that could explain the previous measurement, in the hopes of reducing the error on that query.

To integrate into FEM, we first note that the mechanism only depends on the data through the answers to the workload. If the workload consists of marginal queries, then our methods apply. We derive an expression for the (negative) log-likelihood of the observations, which are the samples from the exponential mechanism in each round, and use this as our objective function for Problem 2. After solving Problem 2 with PROX-PGM, we can use the estimated pseudo-marginals to answer the workload in place of the synthetic dataset generated by FEM.

In this experiment, we use the adult dataset, as that was one of the main datasets considered in FEM. We note that FEM has a number of hyper-parameters, and it is not obvious how to select them, and selecting them incorrectly can result in very poor performance. However, in the authors open source implementation, they provided a set of tuned hyper-parameters a particular dataset/workload pair: the adult dataset and the workload of 64 random 3-way marginals. For a fair comparison, this is the experimental setting we consider.

We run FEM and FEM + Region Graph and note that FEM + Exact failed to run here, because the underlying junction tree necessary to perform exact marginal inference is too large. We report the $L_\infty$ workload error (which is what FEM is designed to minimize), as well as the $L_1$ workload error (which better captures the overall error. The results are shown below. In general, FEM + Region Graph achieves slightly lower error than regular FEM in both $L_\infty$ and $L_1$ error. There is one outlier for $L_\infty$ error when $\epsilon = 1$ that skews the results, and there was negligible improvement at $\epsilon = 0.25$ and $\epsilon = 0.5$ as well. There was consistent improvement in $L_1$ error for every value of $\epsilon$, although the magnitude of the improvement is somewhat small.

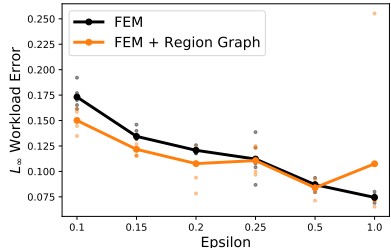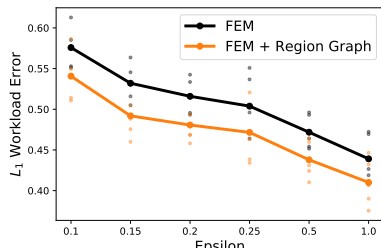

## C.4 Comparison with PriView and Relaxed Projection

As discussed in Section 7, PriView proposed a method for resolving inconsistencies in noisy marginals that can be seen as a less general competitor to us. We compare against that competitor here. We use the implementation of this method available from team DPSyn in the 2018 NIST synthetic data competition [44]. In addition, we compare against a variant of the Relaxed Projection algorithm from [14]. We describe the modifications made to this algorithm in the next section.

To compare these methods with our proposed method, we consider the adult dataset and measure 32 random 2-way marginals using the Gaussian mechanism with privacy parameters $\epsilon \in [0.01, 100]$ and $\delta = 10^{-6}$. In this particular case, PRIVATE-PGM can also run, so we include that as a competitor as well. We report the $L_1$ error of the estimated marginals, averaged over all measured marginals and 5 trials for each method in the table below. All four methods for resolving inconsistencies provide significantly better error than the original noisy marginals.

Ignoring Relaxed Projection, PROX-PGM (Exact) is the best method in every setting except $\epsilon = 100.0$. The second best method is PROX-PGM (Region Graph) in every setting except $\epsilon = 0.01$ and $\epsilon = 100.0$. At the smallest value of $\epsilon$, our method is likely overfitting to the noise, and the estimated pseudo-marginals are likely far from the set of realizable marginals. At the largest value of $\epsilon$, both variants of PROX-PGM simply didn't run for enough iterations (10000 was used in this experiment). Due to the small amount of noise, the true solution to Problem 2 likely does not contain any negatives, and the PriView approach solves the relaxed problem without the non-negativity constraints in closed form. PROX-PGM should eventually converge to the same solution but it would require more than 10000 iterations.

Relaxed Projection (RP) performs slightly better than even PROX-PGM (Exact) for $\epsilon \leq 0.1$, an interesting and surprising observation. We conjecture that this is because RP essentially restricts the search space to distributions which are a mixture of products (as described in the next section). This can be seen as a form of regularization, which can help in the high-privacy / high-noise regime. For $\epsilon > 0.1$, RP is worse than both PROX-PGM (Exact) and PROX-PGM (Region Graph). Moreover, it is the only method whose error does not tend towards 0 as $\epsilon$ gets larger. We suspect this is due to the non-convexity in the problem formulation for RP: it is finding a local minimum to the problem that does not have 0 error. Alternatively, it could be possible that the restircted search space does not include a distribution with near-zero error, although we believe this is a less likely explanation.

| $\epsilon$ | PROX-PGM (Exact) | PROX-PGM (Region Graph) | PriView Consistency | Relaxed Projection | Noisy Marginals |
|---|---|---|---|---|---|
| 0.0100 | 0.4375 $\pm$ 0.0245 | 0.5630 $\pm$ 0.0344 | 0.5229 $\pm$ 0.0202 | 0.4189 $\pm$ 0.0275 | 28.050 $\pm$ 0.1249 |
| 0.0316 | 0.2848 $\pm$ 0.0081 | 0.3277 $\pm$ 0.0100 | 0.3525 $\pm$ 0.0078 | 0.2567 $\pm$ 0.0045 | 8.8782 $\pm$ 0.0254 |
| 0.1000 | 0.1724 $\pm$ 0.0032 | 0.1788 $\pm$ 0.0025 | 0.1965 $\pm$ 0.0051 | 0.1620 $\pm$ 0.0036 | 2.8091 $\pm$ 0.0101 |
| 0.3162 | 0.0908 $\pm$ 0.0009 | 0.0931 $\pm$ 0.0018 | 0.1007 $\pm$ 0.0016 | 0.1031 $\pm$ 0.0025 | 0.8919 $\pm$ 0.0030 |
| 1.0000 | 0.0433 $\pm$ 0.0008 | 0.0447 $\pm$ 0.0006 | 0.0510 $\pm$ 0.0003 | 0.0746 $\pm$ 0.0009 | 0.2853 $\pm$ 0.0007 |
| 3.1622 | 0.0187 $\pm$ 0.0001 | 0.0198 $\pm$ 0.0002 | 0.0229 $\pm$ 0.0003 | 0.0617 $\pm$ 0.0007 | 0.0934 $\pm$ 0.0003 |
| 10.000 | 0.0074 $\pm$ 0.0001 | 0.0087 $\pm$ 0.0001 | 0.0095 $\pm$ 0.0001 | 0.0582 $\pm$ 0.0011 | 0.0324 $\pm$ 0.0001 |
| 31.622 | 0.0037 $\pm$ 0.0000 | 0.0045 $\pm$ 0.0000 | 0.0040 $\pm$ 0.0000 | 0.0579 $\pm$ 0.0017 | 0.0125 $\pm$ 0.0000 |
| 100.00 | 0.0027 $\pm$ 0.0000 | 0.0032 $\pm$ 0.0000 | 0.0018 $\pm$ 0.0000 | 0.0574 $\pm$ 0.0012 | 0.0054 $\pm$ 0.0000 |

### C.5 Implementation Details for Relaxed Projection

The authors of the Relaxed Projection method released their code on GitHub. They provided code to run their end-to-end MWEM-style algorithm, but did not expose the subroutine for performing the relaxed projection in a way that can easily be tested in isolation. For that reason, we compare against a faithful reimplementation of their approach. This reimplementation is available in the open source PRIVATE-PGM repository.

One way to view RP is as optimizing over the set of distributions which are mixtures of products. That is, each row of the relaxed tabualr format can be viewed as a product distribution (if the values for each feature are non-negative and sum to one). For multiple rows, this translates to a format that has capacity to represent a mixture of product distributions. While the authors do not propose restricting the feature values to satisfy the aforementioned constraints, in our reimplemenation, we apply softmax transformations to the table to ensure this invariant holds. This is related to $RAP^{softmax}$ as described by Liu et al. [15], although the interpretation as a mixture of products was not mentioned in that work. For the experiment above, we consider distributions with $100$ mixture components.