# OpenReview forum: "Relaxed Marginal Consistency for Differentially Private Query Answering"
_NeurIPS.cc/2021/Conference — NeurIPS 2021 Poster_

### Official Review · Reviewer_E5iH · 2021-07-16

**Rating:** 6
**Confidence:** 3

**Summary:**

In this paper the authors propose an algorithm for reconstructing data representations from noisy measurements and answering new queries using reconstructed data. Their algorithm, APPGM, builds on a recent method called Private-PGM, a reconstruction method that obtains data representations by solving an optimization problem using graphical models. By relaxing the consistency constraints on overlapping marginals, their algorithm achieves better scalability at the cost of some loss in accuracy.

**Limitations And Societal Impact:**

The authors discussed several limitations when using their method in practice. More discussion on theoretical limitations could be beneficial to potential users of the algorithm.

**Main Review:**

This paper proposes an algorithm for reconstructing data representations from noisy measurements and answering new queries using reconstructed data. Such algorithm is part of the popular select-measure-reconstruct paradigm for answering linear queries on private datasets. In such a framework, data analysts carefully choose a set of queries to send to the data owner. The data owner executes those queries on the private dataset and returns the answers with noise instrumented using standard differential privacy mechanisms to protect the privacy of individuals in the dataset. The data analyst then reconstructs (a representation of) the data and resolves any inconsistencies in the noisy measurements. The reconstructed data can be used to obtain more accurate answers to the original queries or answer new queries with no additional privacy cost.

The proposed APPGM algorithm builds on a recent method called Private-PGM. It has two components, one for reconstructing data and one for answering new queries using reconstructed data. For reconstruction, the authors propose the concept of Region Graph, which leads to pseudo marginals that enforces some but not all internal consistency constraints. The resulting optimization problem can be solved by using existing methods, for example, Prox-PGM with Convex-GBP. For answering new queries, instead of enforcing that the new marginals agree with reconstructed pseudo marginals on overlapping attributes, the authors minimize the L2 norm of constraint violation. By relaxing the consistency constraints, their algorithm achieves better scalability at the cost of some loss in accuracy.


Strength:
+ The concept of Region Graph is novel. It allows various partial consistency constraints to be expressed in a uniform framework.

+ The presentation is clear. The paper did a good job in explaining complex concepts using consistent and sensible mathematical notations.

+ The related work section is comprehensive and includes comparisons to many existing methods in the literature.

Weakness:

- The paper does not contain formal analysis on the loss of accuracy due to relaxation of consistency constraints. As mentioned by the authors, the estimates of out-of-model marginals can be poor in the MWEM experiment. From the graphs, it seems that Private-PGM always outperform APPGM in terms of accuracy when it works. Should one only use APPGM when PPGM fails to run? How large can the drop in accuracy be when both can be applied?

- The paper is not sufficiently self-contained as several important technical details and algorithms are in the supplementary material. As such, it is difficult for reviewers to verify some of the claims in the paper.



Other Comments:
In the experiment (Fig. 2) Private-PGM demonstrates quite different behavior under two clique selection schemes: it can handle many more cliques when they are selected greedily to minimize the junction tree size, compared to when they are selected randomly. Such difference is not present in APPGM. Why is this the case?

Typos:
Line 14: in the design -> in the design of


**Time Spent Reviewing:**

10

---

> ### Author Response · Authors · 2021-08-09
> **Author Response**
>
> We would like to clarify that the concept of a region graph is not new, although the application to this domain and problem is. We included relevant citations to the body of prior work we drew on in Section 4, but should have placed citations with definitions to make this clear. We will fix this in revision.
>
> Regarding the questions -- Private-PGM should always be preferred over APPGM, if it can be run.  However, there are many cases where it simply will not run, in which case APPGM is a good alternative.
>
> We agree it would be nice to have to have a formal analysis on the cost of our approximation. However, very few methods for approximate marginal inference in graphical models are equipped with such guarantees, including Convex-GBP. Instead, these methods are justified as solving a natural relaxation of the free energy problem that underlies exact marginal inference. We will state this as an interesting question to explore in future work.
>
> We were unfortunately limited by space and decided to move the Convex-GBP algorithm to the supplementary material. The precise details of this algorithm are not particularly important for the presentation, only the guarantees it has (Theorem 1). We refer the interested reader to [21] for more information about this approximate marginal inference algorithm.
>
> We will expand the discussion of scalability differences between Private-PGM and APPGM in the discussion of Figure 2c. The reviewer points out an interesting difference between the behavior of Private-PGM and APPGM. Recall that Private-PGM uses an exact Marginal-Oracle, which requires building a junction tree. The junction tree is formed by merging measured cliques into larger super-cliques. If the measured cliques are chosen very carefully, the size of these super-cliques can be kept small, and Private-PGM can scale. However, if they are chosen randomly, their size can quickly grow out of control. APPGM uses a saturated region graph instead of a junction tree, which does not construct any super-cliques. Thus, the performance of APPGM is much less sensitive to the structure of the measured cliques, and mainly only depends on the number of measured cliques. Please refer to the examples in Figure 1 and the supporting text for more information on this matter.

---

### Official Review · Reviewer_8W3Z · 2021-07-17

**Rating:** 6
**Confidence:** 3

**Summary:**

The paper address the problem of answering a complex set of counting queries subject to DP.  They present a basic paradigm that is shared across typical DP approaches: select a set of queries, measure those queries, and reconstruct a dataset from the noisy measurements.  This paper addresses the last phase, where traditional approaches do not scale to large dimensional datasets.  The 2nd phase can easily be done with noise addition mechanisms, e.g. Laplace or Gaussian mechanisms, while the 1st phase is a much more challenging problem that is left open.  The main technical contribution is generalizing Private-PGM (the winning entry in the 2018 NIST DP contest).  Private-PGM suffers from not being scalable in many settings. The new APPGM enjoys many of the benefits of Private-PGM, while also being able to be deployed in more settings.


**Ethical Concerns:**

This paper deals with rigorous notions of privacy.

**Main Review:**

The presentation of Private-PGM could be improved.  It is difficult to see the main challenge with Private-PGM, which is the motivation for this work.  The paper presents the optimization problem and the algorithms very succinctly, but this is done with heavy notation.  In the introduction, the reason for Private-PGM not scaling is due to “the graphical model typically becomes dense”, which is impossible to understand without getting into the details.

The technical contributions in this work seem very broad and not just specific to DP.  Relaxing the consistency constraints from global to local seems natural and might find other applications.  The experiments are also quite interesting.  Although the results show that using Private-PGM can get much better results than the approximate version (with the Region Graph), there gets to be a point where computational resources will not allow for it to get better.  This is where the power of the approach presented in this work comes through because things like MWEM can run much more rounds and get better accuracy given the same computational resources.

Minor Comments:
- What is $s$ and what is P_{s \to r} in line 74?


####
Update: I have reviewed the author responses and other reviewers comments and will keep my score unchanged.

**Time Spent Reviewing:**

2-3 hours

---

> ### Author Response · Authors · 2021-08-09
> **Author Response**
>
> We thank the reviewer for identifying specific places where we can improve our presentation. We plan to add more and clearer discussion of the challenges of scaling Private-PGM to the intro and the beginning of section 3.
>
> We agree that the contributions in this work may be of independent interest outside of the context of differential privacy. In particular, our proposed optimization algorithm (Prox-PGM + Convex-GBP) is a highly scalable algorithm for solving a convex optimization problem over the local marginal polytope. To our knowledge, this is the first algorithm for this general problem that exploits the structure of the constraint set and doesn’t require enumerating the constraints explicitly.

---

### Official Review · Reviewer_85Fd · 2021-07-19

**Rating:** 7
**Confidence:** 4

**Summary:**

For the problem of differentially privacy query answering, a standard paradigm is select-measure-reconstruct. In most methods, the reconstruction step needs to represent the data in vector form, which is impossible when the data is high dimensional. Private-PGM is an approach that uses a graphical model to represent the data more compactly, and it has been shown to improve the scalability of many popular algorithms while decreasing error. However, Private-PGM works best when the graphical model is sparse, and it becomes very inefficient when the graph is dense, which is due to the hardness of exact marginal inference. Therefore, Private-PGM has limited applications for real-world settings. The authors propose a new algorithm that relaxed the marginal inference step to allow Private-PGM to be adapted to more settings. To prove that their method is more scalable, they used the FIRE dataset, which has 15 attributes and 300K rows.


**Limitations And Societal Impact:**

no.

**Main Review:**


This paper provides a technique to make private-pgm more scalable which I believe is a valuable contribution. In particular, they are able to run HDMM with much larger clique feature set without losing much accuracy.  They also show that the their technique can improve scalability of MWEM. However feel that the paper would be stronger if the authors included more comprehensive experiments. Right now the experiments include only one dataset with 15 features.

Minor comments/questions:

Can you obtain better accuracy by running the gradient descent longer or what is the optimization /accuracy trade-off?

**Time Spent Reviewing:**

1

---

> ### Author Response · Authors · 2021-08-09
> **Author Response**
>
> Our main scalability experiment was conducted using synthetic data (Figure 2c), which had 100 attributes and a total domain size of 10^{100}. However, the main scalability challenge is not the dimensionality of the data, but the number and structure of the measurements taken. This was varied in Figure 2c. In fact, Private-PGM has no problem running on the FIRE dataset as long as there are not too many measurements: HDMM+PrivatePGM ran without issue up to workloads of size 16, and MWEM+PrivatePGM ran up to 35-40 rounds. We refer the reviewer to the supplementary material, which contains more experimental results, including some results on five other datasets.
>
> The reviewer raises an interesting question regarding the trade-offs for running the proximal estimation algorithm for more iterations. We have conducted some experiments on this, and generally found that the optimization objective (e.g., distance to noisy marginals) decreases monotonically with more iterations, but the validation accuracy (i.e., distance to true marginals) eventually levels off and stops improving.  We will consider adding a plot to the supplement to demonstrate this behavior.

---

### Official Review · Reviewer_m5oy · 2021-07-19

**Rating:** 7
**Confidence:** 3

**Summary:**

This paper proposes a new algorithm designed to privately and efficiently answer complex sets of counting queries from a database.
The proposed algorithm builds on top of Private-PGM and provides a more scalable way of reconstructing the noisy measurements by relaxing consistency constraints of queries.

**Limitations And Societal Impact:**

The limitations of the proposed algorithm are clearly discussed in the paper

**Main Review:**

- Paper is well written and easy to read
- The experimental results show how the proposed algorithm can improve existing techniques to answer counting queries.
- The authors might consider the local least square approximation inference step in [0] and show how it compares with the proposed algorithm for large number of queries.
- How does the proposed algorithm compares against FEM? In the current paper, the authors add the inference step to FEM, but they do not show a straightforward comparison between FEM and APPGM.


[0] Ryan McKenna, Daniel Sheldon, and Gerome Miklau. Graphical-model based estimation and404
inference for differential privacy. In International Conference on Machine Learning



**Time Spent Reviewing:**

1 hour

---

> ### Author Response · Authors · 2021-08-09
> **Author Response**
>
> Regarding local least squares, note that in Figure 2a, the “Noisy Marginals” baseline is actually equivalent to local least squares, since the measurements are marginals, and the queries are just identity matrices. We will consider adding a local least squares line to Figure 3a, since that was the only known way to run HDMM on domains this large prior to this work. Since local least squares does not enforce inter-marginal consistency, it will yield worse error than the line labeled “HDMM (Expected Error)”.
>
> Since APPGM is not a mechanism, but rather a post-processing tool, we cannot evaluate APPGM by itself. APPGM expects the noisy measurements as input, and they can come from a number of sources: FEM, MWEM, HDMM, and many other randomized mechanisms for discrete data.  Thus, evaluating FEM vs. FEM+APPGM is the most direct comparison possible. While we could have considered an alternative instantiation of APPGM such as HDMM+APPGM, it would no longer be clear how to interpret any performance differences, and isolate whether they are due to APPGM or due to HDMM.

---

### Decision · Program_Chairs · 2021-09-27

**Decision:**

Accept (Poster)

**Comment:**

The reviewers unanimously enjoyed this paper. Thanks for the strong submission.